# CONTROLSPEECH: TOWARDS SIMULTANEOUS ZERO-SHOT SPEAKER CLONING AND ZERO-SHOT LANGUAGE STYLE CONTROL

## ABSTRACT

In this paper, we present **ControlSpeech**, a text-to-speech (TTS) system capable of fully cloning the speaker's voice and enabling arbitrary control and adjustment of speaking style, merely based on a few seconds of audio prompt and a simple textual style description prompt. Prior zero-shot TTS models only mimic the speaker's voice without further control and adjustment capabilities while prior controllable TTS models cannot perform speaker-specific voice generation. Therefore, ControlSpeech focuses on a more challenging task—*a TTS system with controllable timbre, content, and style at the same time*. ControlSpeech takes speech prompts, content prompts, and style prompts as inputs and utilizes bidirectional attention and mask-based parallel decoding to capture codec representations corresponding to timbre, content, and style in a discrete **decoupling codec space**. Moreover, we analyze the many-to-many issue in textual style control and propose the **Style Mixture Semantic Density (SMSD) module**, which is based on Gaussian mixture density networks, to resolve this problem. The SMSD module enhances the fine-grained partitioning and sampling capabilities of style semantic information and enables speech generation with more diverse styles. To facilitate empirical validations, we make available a controllable model toolkit called **ControlToolkit**, which includes all source code, a new style controllable dataset **VccmDataset**, and our replicated competitive baseline models. Our experimental results demonstrate that Control-Speech exhibits comparable or state-of-the-art (SOTA) performance in terms of controllability, timbre similarity, audio quality, robustness, and generalizability. Ablation studies further validate the necessity of each component in ControlSpeech. Audio samples are available at `https://controlspeech.github.io/`.

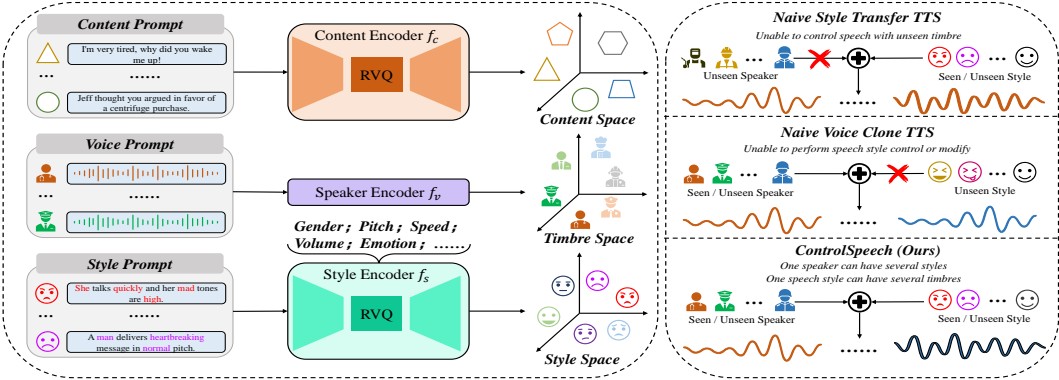

Figure 1: The left panel illustrates the discrete codec representation space of ControlSpeech. The voice prompt, the content description, and the style description correspond to the timbre, content, and style representations in the discrete codec space, respectively. The right panel compares ControlSpeech with previous style-controllable TTS (Naive Style Transfer TTS) and zero-shot TTS systems (Naive Voice Clone TTS). In this comparison, we use the amplitude and frequency of the waveform to represent the style, while the color of the waveform indicates the timbre.

# 1 INTRODUCTION

Over the past decade, the field of speech synthesis has seen remarkable advancements (Ren et al., 2020; Kim et al., 2021; Huang et al., 2022; Ren et al., 2019), achieving synthesized speech that rivals real human speech in terms of expressiveness and naturalness (Tan et al., 2024). Recently, with the development of large language models (Brown et al., 2020; Achiam et al., 2023; Touvron et al., 2023) and generative models in other domains (Ho et al., 2020; Kong et al., 2020; Kim et al., 2020; Lee et al., 2022b), the tasks of zero-shot TTS (Wang et al., 2023; Shen et al., 2023; Le et al., 2023; Jiang et al., 2023c; Borsos et al., 2023) and style-controllable speech synthesis (Guo et al., 2023; Liu et al., 2023; Yang et al., 2023b; Ji et al., 2023) have garnered significant attention in the speech domain due to their powerful zero-shot generation and controllability capabilities. Zero-shot TTS (Wang et al., 2023; Shen et al., 2023; Kharitonov et al., 2023) refers to the ability to perfectly clone an unseen speaker's voice using only a few seconds of a speech prompt, commonly achieved by significantly scaling up both the training data and model sizes. On the other hand, style-controllable TTS (Guo et al., 2023; Yang et al., 2023b) supports the control of a speaker's style (prosody, accent, emotion, etc.) through textual descriptions.

However, these two types of models have their own limitations. As illustrated in the right panel of Figure 1, prior zero-shot TTS (Wang et al., 2023) can clone the voice of any speaker, but the style is fixed and cannot be further controlled or adjusted. Conversely, prior style-controllable TTS (Leng et al., 2023) can synthesize speech in any desired style, but it cannot specify the timbre of the synthesized voice. Although some efforts (Yang et al., 2023b; Liu et al., 2023) have been made to use speaker IDs to control the timbre, these approaches are limited to testing on constrained in-domain datasets and lack the ability to generate audio in the specified style based on the timbre of arbitrary individuals in real-world scenarios. As a result, current speech synthesis systems lack **independent** and flexible control over **content, timbre, and style at the same time**, for example, they are unable to synthesize speech in Trump's voice with a child's joyful style saying "Today is Monday". To address these limitations, we propose a novel model called **ControlSpeech**. To the best of our knowledge, ControlSpeech is the first model to *simultaneously* and *independently* control timbre, content, and style, and demonstrate competitive zero-shot voice cloning and zero-shot style control capabilities.

There are two main challenges to achieve simultaneous control over content, timbre, and style in a TTS system. First, the information from the style prompt and the speech prompt can become entangled and interfere with or contradict each other. For instance, the speech prompt might contain a style different from that described by the textual style prompt; therefore, simply adding a style prompt control module or a speech prompt control module to previous model frameworks (Leng et al., 2023; Wang et al., 2023) is evidently insufficient. Second, there lacks large datasets that fulfill both requirements of zero-shot TTS systems and textual style-controllable TTS systems. Specifically, due to the scarcity of style-descriptive textual data, the training data for mainstream style-controllable TTS systems (Guo et al., 2023; Liu et al., 2023) typically amounts to only a few hundred hours (Ji et al., 2023), far from meeting the requirements of a large-scale, multi-speaker training dataset (Kahn et al., 2020) that is crucial to attain robust zero-shot speaker cloning capabilities. To tackle these two challenges, we explore a novel approach in ControlSpeech that leverages a **pre-trained disentangled representation space** for controllable speech generation. On one hand, disentangling representations enables independent control over content, style, and timbre. On the other hand, utilizing a representation space pre-trained on a large-scale multi-speaker dataset ensures robust zero-shot capabilities of ControlSpeech. In this work, we use the disentangled representation space from (Ju et al., 2024) that is pre-trained on 60,000 hours (Kahn et al., 2020). During the speech synthesis process, we adopt an encoder-decoder architecture (Ren et al., 2020) as the backbone synthesis framework and integrate a high-quality non-autoregressive, confidence-based codec generator (Chang et al., 2022; Borsos et al., 2023; Villegas et al., 2022) as the decoder.

We also identify and analyze the many-to-many issue in textual style-controllable TTS for the first time, that is, different textual style descriptions may correspond to the same audio, **while a single textual style description may be associated with varying degrees of a particular style for the same speaker**. For instance, the phrases "The man speaks at a very rapid pace" and "The man articulates his words with considerable speed" describe the same speech style, yet "The man speaks at a very rapid pace" can also correspond to many audio clips exhibiting **different levels of high speaking rate**. To address this many-to-many issue in style control, we propose a novel module called **Style Mixture Semantic Density Sampling (SMSD)**. This module integrates the global

semantic information of style control and utilizes sampling from a mixed distribution (Zen & Senior, 2014; Hwang et al., 2020) of style descriptions to achieve hierarchical control. Additionally, we incorporate a noise perturbation mechanism within SMSD to further enhance style diversity. The design motivation and detailed architecture of SMSD are elaborated in Section 3.3.

To comprehensively evaluate ControlSpeech's controllability, timbre similarity, audio quality, diversity, and generalization, we create a new dataset called **VccmDataset** based on TextrolSpeech (Ji et al., 2023). Considering the lack of open-source textual style-controllable TTS models, we consolidate our re-implemented competitive baseline models, VccmDataset, and evaluation scripts into a toolkit named **ControlToolkit** and plan to make the toolkit publicly available, to foster advancements in controllable TTS. In summary, our contributions are as follows:

- **Conceptual Contributions.** 1) We conduct detailed analysis of existing zero-shot TTS and style-controllable TTS models and identify their inability to simultaneously and independently control content, style, and timbre in a zero-shot setting. 2) Our findings validate the necessity of disentangling representations of content, style, and timbre and also exploiting pretrained representations to achieve independent control over these speech factors. 3) To the best of our knowledge, this is also the first work to identify and analyze the many-to-many issue in text style-controllable TTS, and propose an effective approach to resolve the issue.
- **Methodological Contributions.** Based on the conceptual contributions summarized above, we propose ControlSpeech, a text-to-speech system capable of independently controlling timbre, style, and content in the zero-shot manner. To tackle the many-to-many problem in style control, we propose a novel Style Mixture Semantic Density (SMSD) module. Furthermore, we investigate integrating various noise perturbation mechanisms within SMSD to enhance control diversity.
- **Experimental Contributions.** We conduct comprehensive experiments and demonstrate that ControlSpeech exhibits comparable or state-of-the-art (SOTA) performance in terms of controllability, timbre similarity, audio quality, robustness, and generalizability. We create a new dataset VccmDataset tailored for style and timbre control, and build ControlToolKit that includes code, VccmDataset, and our replicated competitive baseline models. ControlToolKit will be made publicly available to facilitate fair model comparisons and prompt research in controllable TTS.

## 2 RELATED WORK

In this section, we summarize previous studies on zero-shot TTS, text prompt-based controllable TTS, and discrete codec models related to ControlSpeech and highlight how ControlSpeech differs from them. Detailed discussions and comparisons of related work are in Appendix A.

## 3 CONTROLSPEECH

In this section, we first describe the overall architecture of ControlSpeech in Section 3.1. We then introduce the disentanglement and generation process of the codec in Section 3.2 and the Style Mixture Semantic Density (SMSD) module in Section 3.3. Finally, we present the training loss and inference process of ControlSpeech in Section 3.4.

### 3.1 OVERALL ARCHITECTURE

As illustrated in Figure 2 (a), ControlSpeech is fundamentally an encoder-decoder model (Ren et al., 2020; Ji et al., 2024c) designed for parallel codec generation (Borsos et al., 2023). ControlSpeech employs three separate encoders to encode the input content prompt, style prompt, and speech prompt, respectively. Specifically, the content text is converted into phonemes and fed into the text encoder, while style text is prepended with the special [CLS] token and encoded at the word level using BERT's tokenizer (Devlin et al., 2018). Meanwhile, the speech prompt is processed by the pre-trained codec encoder (Ju et al., 2024) and timbre extractor to capture the timbre information. In Figure 2, the dashed box represents frame-level features, while the solid box represents global features. The Style Mixture Semantic Density (SMSD) module samples style text to generate the corresponding global style representations, which are then combined with text representations from the text encoder via a cross-attention module. The combined representations are then fed into the duration prediction model and subsequently into the codec generator, which is a non-autoregressive Conformer based on

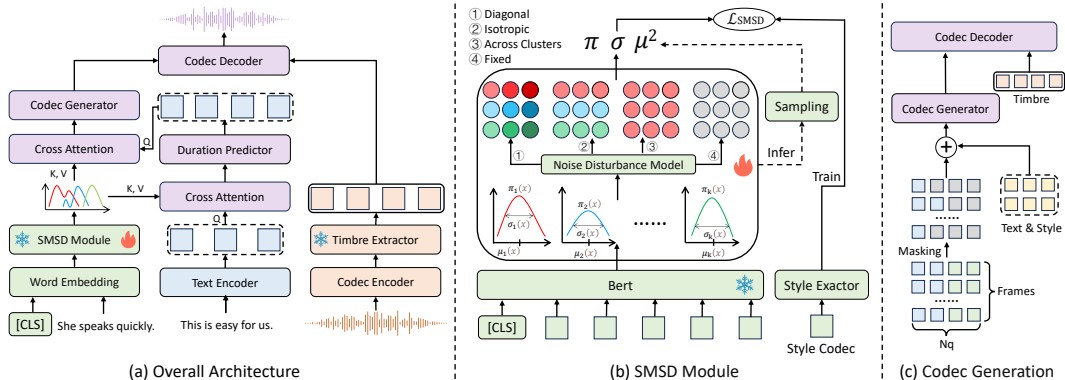

Figure 2: Figure (a) depicts the overall architecture of ControlSpeech, which is an encoder-decoder-based parallel disentangled codec generation model. Figure (b) provides a detailed illustration of the SMSD module in Figure (a), which addresses the many-to-many problem in style control by sampling from the style mixture semantic distribution and incorporating an additional noise perturbator. Figure (c) shows the basic disentanglement process of the codec generator. Through masking, the codec can generate discrete codec representations in a fully non-autoregressive manner.

mask iteration and parallel generation. The timbre extractor is a Transformer encoder that converts the output of the speech encoder into a global vector, representing the timbre attributes. Given the input of a style description $X_s$, a content text $X_c$, and a speech prompt $X_t$, ControlSpeech aims to sequentially generate the corresponding style codec $Y_s$, content codec $Y_c$, and timbre embedding $Y_t$. These representations are then concatenated and upsampled into speech through the pre-trained codec decoder (Ju et al., 2024).

### 3.2 CODEC DECOUPLING AND GENERATION

#### 3.2.1 DECOUPLE CONTENT, STYLE, AND TIMBRE

ControlSpeech leverages the pre-trained disentangled representation space to separate different aspects of speech. We utilize FACodec (Ju et al., 2024) as our codec disentangler and timbre extractor module, since FACodec facilitates codec decoupling and is pre-trained on a large-scale, multi-speaker dataset, ensuring robust zero-shot TTS capabilities. Specifically, during the training process of ControlSpeech, we freeze the corresponding codec encoder to obtain downsampled compressed audio frames $h$ from the target speech $Y$. The frames $h$ are processed through the disentangling quantizer module and the timbre extractor module (Ju et al., 2024) to derive the original content codec $Y_c$, prosody codec $Y_p$, acoustic codec $Y_a$, and timbre information $Y_t$. Theoretically, after excluding the content $Y_c$ and timbre information $Y_t$, the remaining representation collectively is treated as the style codec $Y_s$. In practice, we concatenate the prosody codec $Y_p$ and the acoustic codec $Y_a$ along the channel dimension to obtain the corresponding style codec $Y_s$, as follows:

$$Y_s = concat(Y_p, Y_a) \tag{1}$$

#### 3.2.2 CODEC GENERATION PROCESS

The codec generation process comprises two stages. **In the first stage**, based on the paired text-speech data $\{X, Y_{codec}\}$, where $X = \{x_1, x_2, x_3, \cdots, x_T\}$ represents the cross-attention fusion of the global style representations and the aligned text representations, and $Y_{codec}$ denotes the speech representations through vector quantization, formulated as follows:

$$Y_{codec} = concat(Y_s, Y_c) = C_{1:T,1:N} \in \mathbb{R}^{T \times N} \tag{2}$$

where $T$ denotes the downsampled utterance length, which is equal to the text length extended by the duration predictor. $N$ represents the number of channels for every frame. The row vector of each acoustic code matrix $C_{t,1:N}$ represents the $N$ codes for frame $t$, and the column vector of each acoustic code matrix $C_{1:T,i}$ represents the $i$-th codebook sequence (the length is $T$), where $i \in \{1, 2, \cdots, N\}$.

Following VALL-E (Wang et al., 2023), in the training process of ControlSpeech, we randomly select the $i$-th channel $C_{1:T,i}$ for training. For the generation of the $i$-th channel $P(C_{1:T,i} \mid X_{1:T}; \theta)$, as illustrated in Figure 2 (c), we employ a mask-based generative model as our parallel decoder. We sample the mask $M_i \in \{0, 1\}^T$ according to a cosine schedule (Chang et al., 2022) for codec level $i$, specifically, sampling the masking ratio $p = \cos(u^{'})$ where $u^{'} \sim \mathcal{U}\left[0, \frac{\pi}{2}\right]$. and the mask $M_i \sim Bernoulli(p)$. Here, $M_i$ represents the portion to be masked in the $i$-th level, while $\bar{M}_i$ denotes the unmasked portion in the $i$-th level. As shown in Figure 2 (c), the prediction of this portion $C_{1:T,i}$ is refined based on the prompt $j(j < i)$ channels $C_{1:T,<i}$, and the concatenation of the target text $X_{1:T}$ and the unmasked portion of the $i$-th channel $\bar{M}_i C_{1:T,i}$. Therefore, the prediction for this part can be specified as follows:

$$P(C_{1:T,i} \mid X_{1:T}; \theta) = P(M_i C_{1:T,i} \mid C_{1:T,<i}, X_{1:T}, \bar{M}_i C_{1:T,i}; \theta) \tag{3}$$

**In the second stage**, as illustrated in Figure 2 (c), following AdaSpeech (Chen et al., 2021), we utilize a conditional normalization layer to fuse the previously obtained $Y_{codec}$ and the global timbre embedding $Y_t$, resulting in $Y^{'}$. This result $Y^{'}$ is then processed by the pre-trained codec decoder (Ju et al., 2024) to generate the final speech output $Y$. Specifically, we first use two simple linear layers $W_\gamma$ and $W_\beta$, which take the global timbre embedding $Y_t$ as input and output the scale vectors $\gamma$ and bias vectors $\beta$ respectively. These lightweight, learnable scale vectors $\gamma$ and bias vectors $\beta$ are then fused with $Y_{codec}$. This process can be represented by the following formula:

$$Y = CodecDecoder(W_\gamma Y_t \frac{Y_{codec} - \mu_c}{\sigma_c{}^2} + W_\beta Y_t) \tag{4}$$

where $\mu_c$ and $\sigma_c{}^2$ are the mean and variance of the hidden representation of $Y_{codec}$.

### 3.3 The Style Mixture Semantic Density (SMSD) Module

We identify a **many-to-many** relationship between style text descriptions and their corresponding audio. Specifically, different style descriptions can correspond to the same audio sample (that is, **many-to-one**), while a single style description may correspond to multiple audio samples with varying degrees of the same style (that is, **one-to-many**). More precisely, the many-to-one relationship arises because multiple textual descriptions can refer to the same style of speech. For example, both "Her speaking speed is considerably fast" and "Her speech rate is remarkably fast" can refer to the "fast-speed" speech style and could correspond to the same audio sample. On the other hand, the one-to-many relationship occurs because a single textual description is unable to capture the **varying degrees** of a style. For instance, if we divide the tempo of different speech into 100 levels, any speech with the tempo above 70 may be considered as "fast-speed". As a result, the text description suggesting "fast speed" could correspond to different audio samples with speech rates of 75, 80, or even 90 for the same speaker.

To the best of our knowledge, no mechanisms have been specifically designed to address this one-to-many issue in existing style-controllable TTS systems. It is worth noting that while PromptTTS 2 (Leng et al., 2023) also identifies a one-to-many issue between style descriptions and audio, the one-to-many issue identified in PromptTTS 2 is fundamentally different from the one-to-many issue we identify in ControlSpeech. PromptTTS 2 attributes the one-to-many issue to the absence of the timbre information in the style descriptions, and thus employs a Q-former combined with a diffusion model to generate **the missing latent speech features**. In contrast, we argue that **the textual style descriptions themselves are inherently insufficient to capture the range of variations in one style**, leading to the one-to-many issue.

To address the many-to-many issue in style control, we propose the Style Mixture Semantic Density (SMSD) module. To address the many-to-one issue, similar to previous approaches (Guo et al., 2023; Liu et al., 2023), we utilize a pre-trained BERT model within the SMSD module to extract the semantic representation $X_s^{'}$ from style descriptions, thereby aligning different style texts into the same semantic space and enhancing generalization of out-of-domain style descriptions. To address the one-to-many issue, we observe that **addressing this phenomenon of a single style description corresponding to multiple audio with varying degrees of style closely aligns with the motivation of mixture density networks (MDN)**. We hypothesize that $X_s^{'}$ as the semantic

representation of style can be considered as a global mixture of Gaussian distributions, where different Gaussian distributions represent varying degrees of a particular style. During training, each independent Gaussian distribution is multiplied by a corresponding learnable weight and then summed. By constraining the $KL$ divergence between the style representation distribution of the target audio and the summed mixture density distribution, we establish a one-to-one correspondence between the style text and the target audio. This approach also enhances the diversity of style control directly with the text descriptions. During inference, we sample from the mixture of style semantic distributions to obtain an independent Gaussian distribution, with each sampled distribution reflecting different degrees of the same style. Additionally, to further enhance the diversity of style control, we incorporate a noise perturbation module within the MDN network of SMSD in ControlSpeech. **The noise perturbation module controls the isotropy of perturbations across different dimensions.**

Specifically, one raw style prompt $X_s = [X_1, X_2, X_3, \cdots, X_L]$ is prepended with a $[CLS]$ token, then converted into word embedding, and fed into the BERT model, where $L$ denotes the length of the style prompt. The hidden vector corresponding to the $[CLS]$ token is regarded as the global style semantic representation $X_s^{'}$, which guides generation and sampling of subsequent modules.

Based on the MDN network (Zen & Senior, 2014; Duan, 2019; Du & Yu, 2021), we aim to regress the target style representation $Y_s^{'} \in \mathbb{R}^d$, using the style semantic input representation $X_s^{'} \in \mathbb{R}^n$ as covariates, where $d$ and $n$ are the respective dimensions. We model the conditional distribution as a mixture of Gaussian distribution, as follows:

$$P_\theta(Y_s^{'}|X_s^{'}) = \sum_{k=1}^{K} \pi_k \mathcal{N}(\mu^{(k)}, \sigma^{2(k)}) \tag{5}$$

where $K$ is a hyperparameter as the number of independent Gaussian distribution, and other mixture distribution parameters $\pi_k, \mu^k, \sigma^{2(k)}$ are output of a neural MDN network $f_\theta$ based on the input style semantic representation $X_s^{'}$, as follows:

$$\pi \in \Delta^{K-1}, \mu^{(k)} \in \mathbb{R}^d, \sigma^{2(k)} \in S_+^d = f_\theta(X_s^{'}) \tag{6}$$

Note that the sum of the mixture weights is constrained to 1 during the training phase, which is achieved by applying a softmax function on the corresponding neural network output $\alpha_k$, as follows:

$$\pi_k = \frac{exp(a_k)}{\sum_{k=1}^{K} exp(a_k)} \tag{7}$$

To further enhance the diversity of style control, we design a specialized noise perturbation module within the SMSD module to constrain the noise model. As illustrated by the circles within the SMSD module in Figure 2 (b), this noise perturbation module regulates the isotropy of perturbations $\varepsilon$ across different dimensions in variance $\sigma^{2(k)}$. The four types of perturbations from left to right in Figure 2 (b) are as follows:

- **Fully factored**: $\sigma^{2(k)} = f_\theta(X_s^{'}) + f_\theta(\varepsilon) = diag(\sigma^{2(k)}) \in \mathbb{R}_+^d$, which predicts the noise level for each dimension separately.
- **Isotropic**: $\sigma^{2(k)} = f_\theta(X_s^{'}) + f_\theta(\varepsilon) = \sigma^{2(k)} I \in \mathbb{R}_+$, which assumes the same noise level for each dimension over $d$.
- **Isotropic across clusters**: $\sigma^{2(k)} = f_\theta(X_s^{'}) + f_\theta(\varepsilon) = \sigma^2 I \in \mathbb{R}_+$, which assumes the same noise level for each dimension over $d$ and cluster.
- **Fixed isotropic** is the same as Isotropic across clusters but does not learn $\sigma^2$.

As shown in the experimental results in Appendix I, *isotropic across clusters* outperforms the other types for striking a balance between accuracy and diversity and is used as the mode for noise perturbation. We obtain more robust mean, variance, and weight parameters for the mixture of Gaussian distributions with the noise perturbation module. The training objective of the SMSD module is the negative log-likelihood of the observation $Y_s^{'}$ given its input $X_s^{'}$. The loss function is formulated as follows. Details for deriving the non-convex $\mathcal{L}_{SMSD}$ are in Appendix C.

$$\mathcal{L}_{SMSD} = -logP_\theta(Y_s^{'}|X_s^{'})$$

$$\propto -\sum_{k=1}^{K}(\pi_k exp(-\frac{1}{2}(Y_s^{'} - \mu^{(k)})^T \sigma^{2(k)^{-1}}(Y_s^{'} - \mu^{(k)}) - \frac{1}{2}logdet\sigma^{2(k)}))$$

$$= -logsumexp_k(log\pi_k - \frac{1}{2}(Y_s^{'} - \mu^{(k)})^T \sigma^{2(k)^{-1}}(Y_s^{'} - \mu^{(k)}) - \frac{1}{2}logdet\sigma^{2(k)}) \quad (8)$$

$$= -logsumexp_k(log\pi_k - \frac{1}{2}\left\|\frac{Y_s^{'} - \mu^{(k)}}{\sigma}\right\|^2)$$

### 3.4 TRAINING AND INFERENCE

During the training process, the duration predictor is optimized using the mean square error loss, with the extracted duration serving as the training target. We employ the Montreal Forced Alignment (MFA) tool (McAuliffe et al., 2017) to extract phoneme durations, and denote the loss for the duration predictor as $\mathcal{L}_{dur}$. The codec generator module is optimized using cross-entropy loss. We randomly select a channel for optimization and denote this loss as $\mathcal{L}_{codec}$. In the SMSD module, the target style representation $Y_s^{'}$ is the global style representation obtained by passing style codec $Y_s$ through the style extractor. During training, we feed the ground truth style representation $Y_s^{'}$ and the ground truth duration into the codec generator and duration predictor, respectively. The overall loss $\mathcal{L}$ for ControlSpeech is the sum of all these losses:

$$\mathcal{L} = \mathcal{L}_{codec} + \mathcal{L}_{dur} + \mathcal{L}_{SMSD} \quad (9)$$

During the inference stage, we initiate the process by inputting the original stylistic descriptor $X_s$ into the BERT module to obtain the style semantic representation $X_s^{'}$, and then input $X_s^{'}$ into the SMSD module to obtain the corresponding $\pi$, $\mu$ and $\sigma^2$. By directly sampling $X_s^{'}$, we can derive the predicted style distribution. Subsequently, we iteratively generate discrete acoustic tokens by incorporating the predicted style into the text state and employing the confidence based sampling scheme (Chang et al., 2022; Borsos et al., 2023). Specifically, we perform multiple forward passes, and at each iteration $j$, we sample candidates for the masked positions. We then retain $P_j$ candidates based on their confidence scores, where $P_j$ follows a cosine schedule. Finally, by integrating the timbre prompt through the condition normalization layer and feeding it into the codec decoder, we generate the final speech output.

## 4 EXPERIMENTS

### 4.1 EXPERIMENTAL SETUP

**VccmDataset.** To the best of our knowledge, there is no large-scale TTS dataset that includes both text style prompts and speaker prompts. We build upon the TextrolSpeech dataset (Ji et al., 2023) and create **VccmDataset**. TextrolSpeech comprises a total of 330 hours of speech data along with 236,203 style description texts. Based on TextrolSpeech, we optimize the pitch distribution, label boundaries, the dataset splits, and then select new test sets. Specifically, we use LibriTTS and the emotional data from TextrolSpeech as the base databases, and annotate each speech sample with five attribute labels: gender, volume, speed, pitch, and emotion. We use the gender labels available in the online metadata. Regarding volume, we compute the L2-norm of the amplitude of each short-time Fourier transform (STFT) frame. We utilize the Montreal forced alignment (MFA) tool (McAuliffe et al., 2017) to extract phoneme durations and silence segments. Subsequently, we calculate the average duration of each phoneme within voiced segments for the speaking speed. The Parselmouth 3 tool [1] is employed to extract fundamental frequency (f0) and calculate the geometric mean across all voiced regions as pitch values. After obtaining the speed, pitch, and volume values, we partition speech samples into 3 categories (high/normal/low) according to the proportion of speed, pitch, and volume values

---

[1] https://github.com/YannickJadoul/Parselmouth

Table 1: The **style controllability** evaluation results of style-controlled models on VccmDataset *test set A*. *Pitch*, *Speed*, *Volume*, *Emotion* denote accuracy of the style. $\pm$ denotes standard deviation.

| Model | Clone Timbre | Control Style | Pitch ↑ | Speed ↑ | Volume ↑ | Emotion ↑ | WER ↓ | Spk-sv ↑ | MOS-Q ↑ |
|---|---|---|---|---|---|---|---|---|---|
| GT Codec | - | - | 0.954 | 0.885 | 0.977 | 0.758 | 2.6 | 0.96 | 4.25±0.10 |
| Salle | × | √ | 0.788 | 0.756 | 0.831 | 0.389 | 5.5 | - | 3.52±0.14 |
| PromptStyle | × | √ | 0.831 | 0.786 | 0.787 | 0.366 | 3.3 | 0.84 | 3.74±0.11 |
| InstructTTS | × | √ | 0.849 | 0.761 | 0.822 | 0.412 | 3.0 | 0.86 | 3.81±0.12 |
| PromptTTS 2 | × | √ | **0.867** | 0.785 | 0.825 | 0.406 | 3.1 | - | 3.83±0.11 |
| ControlSpeech (**Ours**) | √ | √ | 0.833 | **0.829** | **0.894** | **0.557** | **2.9** | 0.89 | **3.91±0.09** |

respectively. Considering the close proximity of attribute values of speech samples between adjacent categories, we exclude the 5% of data samples at the boundaries of each interval for each attribute. This ensures greater distinctiveness for each label. Particularly, due to the significant difference in the pitch distribution between male and female voices, we use gender-specific thresholds to bin the pitch into three different levels. After obtaining more accurate labels through these procedures, we align each audio segment with the corresponding style description text in TextrolSpeech based on the labeled attributes to obtain the VccmDataset. We then select four distinct test sets from VccmDataset, namely, *test set A*, *test set B*, *test set C*, *test set D*, for comprehensively evaluating the performance of ControlSpeech on controllable tasks. Details of the VccmDataset test sets are in Appendix D.

**ControlToolkit and Baselines.** To prompt research in the controllable TTS field, we build **ControlToolkit**. ControlToolKit provides complete download links for VccmDataset. There is a notable lack of open-source models in the field of controllable speech synthesis. To ensure a fair comparison of the actual performance of various models, we reimplement several SOTA style-controllable models, including PromptStyle (Liu et al., 2023), Salle (Ji et al., 2023), InstructTTS (Yang et al., 2023b), and PromptTTS 2 (Leng et al., 2023), to serve as primary comparative models for evaluating the controllability of ControlSpeech. For the comparison of voice cloning effectiveness, we reimplement the VALL-E model (Wang et al., 2023) and the MobileSpeech model (Ji et al., 2024c), which are representatives of the autoregressive paradigm and the parallel generation paradigm, respectively. ControlToolKit integrates these reimplemented baseline models together with comprehensive training and inference interfaces for them, along with pre-trained model weights, all source code, and testing scripts. All components in ControlToolKit will be made publicly available.

**Evaluation Metrics and Experimental Settings.** For **objective evaluations**, we adopt the common metrics used in prior works (Guo et al., 2023; Ji et al., 2023; Leng et al., 2023). To evaluate the model's style controllability, we use **accuracy** of pitch, speaking speed, volume, emotion as the metrics, which measures the correspondence between the style factors in the output speech and those in the prompts. We evaluate timbre similarity (**Spk-sv**) between the original prompt and the synthesized speech, and evaluate speech synthesis accuracy and robustness by using an ASR system to transcribe the synthesized speech and computing word error rate (**WER**) against the content prompt. For **subjective evaluations**, we conduct mean opinion score (MOS) evaluations on the test set to measure audio naturalness via crowdsourcing. We further analyze MOS in two aspects: MOS-Q (Quality, assessing clarity and naturalness of the duration and pitch) and MOS-S (Speaker similarity). We also design new subjective MOS metrics: **MOS-TS** (Timbre similarity), **MOS-SD** (Style diversity), and **MOS-SA** (Style accuracy). Details of the evaluation metrics, experimental settings, and specifics of model architecture are provided in Appendix E, F, and G, respectively.

### 4.2 RESULTS AND DISCUSSIONS

**The GT Codec model in all tables denotes synthesizing speech using ground truth codecs of test samples with FACodec.** In each table, best results for each metric, excluding GT Codec, are in bold.

**Evaluation on style controllability.** We first compare the performance of ControlSpeech with various SOTA models on the style controllability task. The evaluation is conducted on the 1,500-sample VccmDataset *test set A*. To eliminate the influence of timbre variations on the controllability results of ControlSpeech, we use the ground truth (GT) timbre as the prompt for ControlSpeech. We compare the controllability of the models using pitch accuracy, speed accuracy, volume accuracy, and emotion accuracy. Additionally, we measure the audio quality generated by the models using

WER, timbre similarity (Spk-sv), and MOS-Q (Mean Opinion Score for Quality). Results are shown in Table 1, and we drew the following conclusions:

**1)** GT Codec exhibits high reconstruction quality. However, it shows limitations in emotion and speech speed classification accuracy. We attribute this to accumulated errors introduced by the test model. Additionally, the emotion classification experiment does not include the neutral emotion classification results, which better highlights the model's emotion control capabilities but also presents a more challenging task for all models. **2)** Comparing ControlSpeech with other baselines on controllability metrics, we find that, except for pitch accuracy, **ControlSpeech achieves best results in volume, speed, and emotion classification accuracy**. Upon analyzing the synthesized audio of ControlSpeech, we attribute the degraded pitch accuracy to the difficulty arising from simultaneously controlling different timbres and styles. **3)** In terms of Spk-sv, MOS-Q, and WER metrics, **the audio generated by ControlSpeech demonstrates best timbre similarity, audio quality, and robustness**.

Table 2: The **timbre cloning** results of different zero-shot models on the VccmDataset **test set B**. None of the speakers appear in the training set.

| Model | Clone Timbre | Control Style | WER ↓ | MOS-Q ↑ | MOS-S ↑ |
|---|---|---|---|---|---|
| GT Codec | - | - | 2.3 | 4.21±0.14 | 4.29±0.12 |
| VALL-E | √ | × | 6.7 | 3.76±0.13 | 3.89±0.13 |
| MobileSpeech | √ | × | 4.1 | 3.94±0.09 | **4.01±0.11** |
| ControlSpeech (**Ours**) | √ | √ | **3.3** | **3.95±0.12** | 3.96±0.14 |

**Evaluation on the timbre cloning task.**  To evaluate the timbre cloning capability of ControlSpeech in an out-of-domain speaker scenario, we compare the performance of ControlSpeech with SOTA models such as VALL-E and MobileSpeech on the out-of-domain speaker test set (**test set B**) from the VccmDataset. The **test set B** consists of 1,086 test utterances, and we ensure that none of the speakers in test set B appear in the training set. To ensure a fair comparison, both VALL-E and MobileSpeech are retrained using the VccmDataset training set. The experimental results are shown in Table 2. We observe that **in terms of the robustness metric (WER), the zero-shot TTS systems that are trained on small datasets perform worse than ControlSpeech**. We attribute these performance gains of ControlSpeech to its pre-trained speaker prompt component, which is trained on large-scale, 60,000 hours of multi-speaker data. Additionally, in terms of the MOS-Q (Quality) and MOS-S (Speaker similarity) metrics, we find that **on top of its style control capabilities, ControlSpeech also maintains performance comparable to zero-shot TTS systems on the timbre cloning task**.

**Evaluation on the out-of-domain style control task.**  We further evaluate the controllability of style-controllable models with out-of-domain style descriptions. We compare the performance of ControlSpeech with controllable baseline models on the VccmDataset **test set C**. The **test set C** comprises 100 test utterances, *with style prompts rewritten by experts*. None of the test set style prompts are present in the training set. Results are shown in Table 3. We find that **the generalization performance of ControlSpeech is remarkably better than that of the baseline models**, which could be attributed to the SMSD module and its underlying mixture density network mechanism. The accuracies of speech speed and volume from ControlSpeech are markedly better than those from baseline models, especially in terms of the volume accuracy. ControlSpeech also yields best WER, MOS-Q, and speaker timbre similarity. Similar to the results shown in Table 1, the pitch accuracy of ControlSpeech is slightly lower. We believe this is due to pitch inconsistencies arising from the simultaneous control of style and timbre cloning. Note that there is no significant difference between the **test set A** and **test set C**, except the style descriptions in **test set C** are out-of-domain while those in **test set A** are in-domain. Comparing Table 3 and Table 1, degradations from ControlSpeech on all metrics are much smaller than degradations from baselines.

**Evaluation on addressing the many-to-many issue.**  To better evaluate the performance of style-controllable models on addressing the many-to-many issue, we compare ControlSpeech with controllable baseline models on the VccmDataset **test set D**. Results are shown in Table 4. We find that **ControlSpeech markedly outperforms PromptStyle and InstructTTS on both MOS-SA (style accuracy) and MOS-SD (style diversity) metrics**. This suggests that the unique SMSD module in ControlSpeech enables the model to synthesize both *accurate* and *diverse* speech.

Table 3: The **out-of-domain style control** results of different style-controlled models on the Vccm-Dataset *test set C*. None of the style prompts are present in the training set.

| Model | Pitch ↑ | Speed ↑ | Volume ↑ | WER ↓ | Spk-sv ↑ | MOS-Q ↑ |
|---|---|---|---|---|---|---|
| GT Codec | 0.85 | 0.87 | 0.91 | 2.8 | 0.96 | 4.25±0.11 |
| Salle | 0.67 | 0.55 | 0.56 | 6.4 | - | 3.47±0.08 |
| PromptStyle | **0.77** | 0.57 | 0.49 | 3.7 | 0.81 | 3.65±0.11 |
| InstructTTS | 0.75 | 0.55 | 0.54 | 3.1 | 0.82 | 3.76±0.14 |
| PromptTTS 2 | 0.76 | 0.59 | 0.58 | 3.3 | - | 3.54±0.13 |
| ControlSpeech (**Ours**) | 0.75 | **0.73** | **0.85** | **3.0** | **0.88** | **3.86±0.12** |

Table 4: The results under **many-to-many style control conditions** on VccmDataset *test set D*. MOS-TS, MOS-SA, MOS-SD measure timbre stability, accuracy and diversity of style generation.

| Model | MOS-TS ↑ | MOS-SA ↑ | MOS-SD↑ |
|---|---|---|---|
| PromptStyle | 3.81±0.10 | 3.45±0.13 | 3.53±0.12 |
| InstructTTS | 3.89±0.12 | 3.57±0.11 | 3.48±0.14 |
| ControlSpeech w/o SMSD | 3.95±0.08 | 3.59±0.09 | 3.66±0.11 |
| ControlSpeech | **4.01±0.10** | **3.84±0.12** | **4.05±0.09** |

## 4.3 ABLATION STUDIES

We validate the necessity of the codec decoupling scheme and the SMSD module. We also investigate the impact of hyperparameters for mixed distributions and various noise models in Appendix H and I.

Table 5: An ablation experiment on impact of **codec decoupling** on the VccmDataset *test set A*.

| Model | Pitch ↑ | Speed ↑ | Volume ↑ | Emotion ↑ |
|---|---|---|---|---|
| ControlSpeech w/o decoupling | 0.492 | 0.517 | 0.582 | 0.237 |
| ControlSpeech | **0.833** | **0.829** | **0.894** | **0.557** |

**Decouple codec.** To analyze the impact of decoupling codec, we maintain the main framework of ControlSpeech but use a non-decoupled Encodec to represent discrete audio in the TTS model. Furthermore, during training, we keep the text encoder, duration predictor, and codec generator in ControlSpeech unchanged; however, we directly encode the speech prompt and style prompt using the speech encoder and style encoder (replicated from the structure of the text encoder) respectively, then feed them into the codec generator through cross attention. We denote this model as **ControlSpeech w/o decoupling** and evaluate it using the prompt version of VccmDataset *test set A*. As shown in Table 5, ControlSpeech w/o decoupling performs substantially worse in controllability compared to ControlSpeech, suggesting that the speech prompt and style prompt indeed may interfere with each other, making it difficult to simultaneously clone timbre and control style with this naive approach.

**The SMSD module.** We evaluate the effectiveness of the SMSD module in addressing the many-to-many style control problem. Specifically, we replace the SMSD module with a style encoder (replicated from the structure of the text encoder) and denote this model as **ControlSpeech w/o SMSD**. As shown in Table 4, ControlSpeech w/o SMSD performs markedly worse in terms of MOS-SA and MOS-SD compared to ControlSpeech, which strongly validates that the SMSD module enables more fine-grained control of the model's style and increases style diversity through style sampling. We visualize the distribution of the SMSD mixed density network under varying pitch/speed/volume (details in Appendix B). Our results verify that SMSD effectively distinguishes between different types of styles and the style control module exhibits substantial diversity.

## 5 CONCLUSION

We present ControlSpeech, the first TTS system capable of simultaneously performing zero-shot timbre cloning and zero-shot style control. We disentangle style, content, and timbre, and generate the corresponding codec representations through a non-autoregressive, mask-based iterative codec generator. Additionally, we identify a many-to-many problem in style control and design a unique Style Mixed Semantic Density module to mitigate this issue. The limitation and future work of ControlSpeech are discussed in Appendix J.

## 6 ETHICS STATEMENT

ControlSpeech is capable of zero-shot voice cloning; hence, there are potential risks from misuse, such as voice spoofing. For any real-world applications involving unseen speakers, it is crucial to establish protocols ensuring the speaker's authorization over using the certain speaker's voice. Also, to mitigate these risks, we will also develop approaches such as speech watermarking technology to identify whether a given audio is synthesized by ControlSpeech.

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

# A RELATED WORK

## A.1 TEXT PROMPT BASED CONTROLLABLE TTS

Some recent studies propose to control speech style through natural language prompts, providing a more interpretable and user-friendly approach for style control. PromptTTS (Guo et al., 2023) employs manually annotated text prompts to describe four to five attributes of speech (gender, pitch, speaking speed, volume, and emotion) and trains model on LibriTTS and two synthesized speaker datasets. InstructTTS (Yang et al., 2023b) employs a three-stage training approach to capture semantic information from natural language style prompts as conditioning to the TTS system. Textrolspeech (Ji et al., 2023) introduces an efficient architecture which treats textual controllable TTS as a language model task. PromptStyle (Liu et al., 2023) proposes a two-stage TTS approach for cross-speaker style transfer with natural language descriptions based on VITS (Kim et al., 2021). PromptTTS 2 (Leng et al., 2023) proposes an automatic description creation pipeline leveraging large language models (LLMs) (Bubeck et al., 2023) and adopts a diffusion model to capture the one-to-many relationship. Audiobox (Vyas et al., 2023) propose a unified model based on flow-matching that is capable of generating and controlling various audio modalities. However, regarding the speech modality, while AudioBox supports multiple inputs, it does not decouple the speech prompt from the style prompt. Consequently, when there is a conflict between the styles in the speech prompt and the style text prompt, it significantly impacts the controllability. We also validate the necessity of decoupling in our ablation study presented in Table 5. **It is noteworthy that existing style-controllable TTS models are either speaker-independent or can only control timbre using speaker IDs, without the capability for timbre cloning**. The introduction of ControlSpeech expands the scope of the controllable TTS task. Furthermore, to the best of our knowledge, **ControlSpeech is the first model to identify the many-to-many problem in the field of style control and we proposes a novel SMSD module to address this issue**.

## A.2 ACOUSTIC CODEC MODELS

In recent times, neural acoustic codecs (Zeghidour et al., 2021; Défossez et al., 2022; Kumar et al., 2024) have demonstrated remarkable capabilities in reconstructing high-quality audio at low bitrates. Typically, these methods employ an encoder to extract deep features in a latent space, which are subsequently quantized before being fed into the decoder. To elaborate, Soundstream (Zeghidour et al., 2021) utilizes a model architecture comprising a fully convolutional encoder/decoder network and a residual vector quantizer (RVQ) to effectively compress speech. Encodec (Défossez et al., 2022) employs a streaming encoder-decoder architecture with a quantized latent space, trained in an end-to-end fashion. AudioDec (Wu et al., 2023) has demonstrated the importance of discriminators. PromptCodec (Pan et al., 2024) enhances representation capabilities through additional input prompts. DAC (Kumar et al., 2024) significantly improves reconstruction quality through techniques like quantizer dropout and a multi-scale STFT-based discriminator. Vocos (Siuzdak, 2023) eliminates codec noise artifacts using a pre-trained Encodec with an inverse Fourier transform vocoder. HILCodec (Ahn et al., 2024) introduces the MFBD discriminator to guide codec modeling. APCodec (Ahn et al., 2024) further enhances reconstruction quality by incorporating ConvNextV2 modules in the encoder and decoder. HiFi-Codec (Yang et al., 2023a) proposes a parallel GRVQ structure, achieving good speech reconstruction with just four quantizers. Language-Codec (Ji et al., 2024a) introduces the MCRVQ mechanism to evenly distribute information across the first quantizer, also requiring only four quantizers for excellent performance across various generative models. Single-Codec (Li et al., 2024) designs additional BLSTM, hybrid sampling, and resampling modules to ensure basic performance with a single quantizer, though reconstruction quality still needs improvement. TiCodec (Ren et al., 2024) models codec space by distinguishing between time-independent and time-dependent information. FACodec (Ju et al., 2024) further decouples codec space into content, style, and acoustic detail modules. Additionally, recognizing the importance of semantic information in generative models, recent efforts have begun integrating semantic information into codec models. RepCodec (Huang et al., 2024) learns a vector quantization codebook by reconstructing speech representations from speech encoders like HuBERT. SpeechTokenizer (Zhang et al., 2023) enriches the semantic content of the first quantizer through semantic distillation. FunCodec (Du et al., 2024) makes semantic tokens optional and explores different combinations. SemanticCodec (Liu et al., 2024) is based on quantized semantic tokens and further reconstructs acoustic information using an audio encoder and diffusion model. WavTokenizer (Ji et al., 2024b) represents the latest

state-of-the-art codec model, capable of reconstructing high-quality audio using only forty discrete codebooks. **Given that ControlSpeech requires disentangled discrete audio representations that are pre-trained on large-scale multi-speaker data, we select FACodec (Ju et al., 2024) as the tokenizer for ControlSpeech**.

### A.3 ZERO-SHOT TTS

Zero-shot speech synthesis refers to the ability to synthesize the voice of an unseen speaker based solely on a few seconds of audio prompt, also known as voice cloning. In recent months, with the advancement of generative large-scale models, a plethora of outstanding works have emerged. VALL-E (Wang et al., 2023) leverages discrete codec representations and combines autoregressive and non-autoregressive models in a cascaded manner, preserving the powerful contextual capabilities of language models. NaturalSpeech 2 (Shen et al., 2023) employs continuous vectors instead of discrete neural codec tokens and introduces in-context learning to a latent diffusion model. NaturalSpeech 3 (Ju et al., 2024) proposes a TTS system with novel factorized diffusion models to generate natural speech in a zero-shot way. SpearTTS (Kharitonov et al., 2023) and Make-a-Voice (Huang et al., 2023) utilize semantic tokens to reduce the gap between text and acoustic features. VoiceBox (Le et al., 2023) is a non-autoregressive flow-matching model trained to infill speech, given audio context and text. Mega-TTS (Jiang et al., 2023c;b;a), on the other hand, utilizes traditional mel-spectrograms, decoupling timbre and prosody and further modeling the prosody using an autoregressive approach. VoiceBox (Le et al., 2023) and P-flow (Kim et al., 2024) employ flowing models as generators, demonstrating robust generative performance. SoundStorm (Borsos et al., 2023) and MobileSpeech (Ji et al., 2024c) utilize a non-autoregressive and mask-based iterative generation method, achieving an excellent balance between inference speed and generation quality. **It is noteworthy that existing zero-shot TTS models are unable to achieve arbitrary language style control. ControlSpeech is the first TTS model capable of simultaneously performing zero-shot timbre cloning and style control.**

## B   DISTRIBUTION VISUALIZATION

In this section, we visualize the distribution of the SMSD mixed density network. As shown in Figure 3, we select the original style descriptions from TextrolSpeech and visualize the distributions produced by the SMSD module under three experimental settings: varying pitch (high/low), speech rate (fast/slow), and volume (high/low). Each experimental setting includes 1,000 different style descriptions, with other factors held constant. For example, in the speech rate experiment, both pitch and volume descriptions are set to "normal." We employ t-SNE for dimensionality reduction of the features. Our results show that the SMSD module effectively distinguishes between different types of styles, and the mixed density distribution is not confined to a small region, indicating that the style control module exhibits substantial diversity.

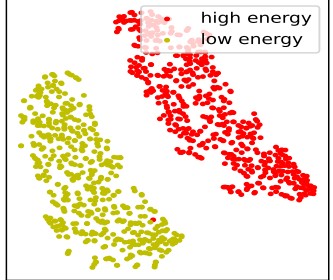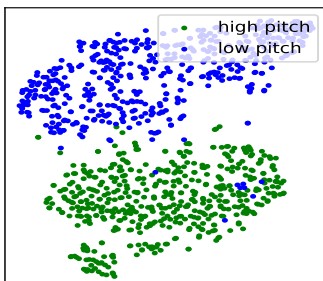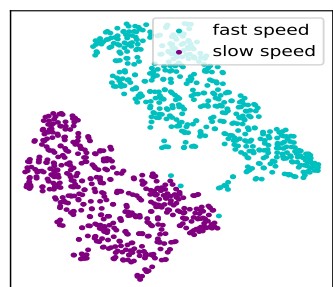

Figure 3: The t-SNE visualization of mixture density distribution after the SMSD module.

## C  THE SMSD LOSS

The loss function for the SMSD module represents the conditional probability of the input style representation $X_s'$ given the target global style $Y_s'$. We further refine this into a maximum likelihood loss involving the style distribution parameters $\pi_k$, $\mu^{(k)}$, $\sigma^{2(k)}$ derived through the MDN network and noise perturbation module. The detailed derivation of the loss function is as follows.

$$
\begin{aligned}
\mathcal{L}_{SMSD} &= -logP_\theta(Y_s'|X_s') \\
&\propto -\sum_{k=1}^{K}(\pi_k exp(-\frac{1}{2}(Y_s'-\mu^{(k)})^T\sigma^{2(k)^{-1}}(Y_s'-\mu^{(k)}) - \frac{1}{2}logdet\sigma^{2(k)})) \\
&= -logsumexp_k(log\pi_k - \frac{1}{2}(Y_s'-\mu^{(k)})^T\sigma^{2(k)^{-1}}(Y_s'-\mu^{(k)}) - \frac{1}{2}logdet\sigma^{2(k)}) \\
&= -logsumexp_k(log\pi_k - \frac{1}{2}\left\|\frac{Y_s'-\mu^{(k)}}{\sigma^{(k)}}\right\|^2 - \left\|log(\sigma^{(k)}))\right\|_1) \\
&= -logsumexp_k(log\pi_k - \frac{1}{2}\left\|\frac{Y_s'-\mu^{(k)}}{\sigma^{(k)}}\right\|^2 - dlog(\sigma^{(k)})) \\
&= -logsumexp_k(log\pi_k - \frac{1}{2}\left\|\frac{Y_s'-\mu^{(k)}}{\sigma^{(k)}}\right\|^2 - dlog(\sigma)) \\
&= -logsumexp_k(log\pi_k - \frac{1}{2}\left\|\frac{Y_s'-\mu^{(k)}}{\sigma}\right\|^2)
\end{aligned}
\tag{10}
$$

## D  VCCMDATASET TEST SET

To further validate ControlSpeech's ability to simultaneously control style and clone speaker timbre, we create four types of test sets in the VccmDataset: the main test set (***test set A***), the out-of-domain speaker test set (***test set B***), the out-of-domain style test set (***test set C***), and the special case test set (***test set D***). Each test set corresponds to each experiment in Section 4: style controllability experiments, out-of-domain speaker cloning experiments, out-of-domain style controllability experiments, and many-to-many style control experiments, respectively. We randomly select 1,500 audio samples as the ControlSpeech main test set (***test set A***) and match the corresponding prompt voice based on speaker IDs. Additionally, to evaluate ControlSpeech's performance on out-of-domain timbre and styles, we further filter an appropriate test set (speakers that are not present in the training set) and enlist language experts to compose style descriptions distinct from those in TextrolSpeech. Using these two methods, we generate the out-of-domain speaker test set (***test set B***) and the out-of-domain style test set (***test set C***). The special case test set (***test set D***) is designed to evaluate the model's performance under many-to-many style control conditions. Firstly, we select four groups of speakers, each of whom is matched with 60 different style descriptions while the content text remains fixed. This particular set of test samples is referred to as ***test set D1***. We further select six distinct style descriptions paired with 50 different timbre prompts, with pitch, speed, and volume labels set to the following combinations: normal, fast, normal; normal, slow, normal; high, normal, normal; low, normal, normal; normal, normal, high; and normal, normal, low, respectively. This set of special test samples is referred to as ***test set D2***.

## E  EVALUATION METRICS

For objective evaluations, we adopt the metrics used in prior works (Guo et al., 2023; Ji et al., 2023; Leng et al., 2023). To evaluate the model's style controllability, we use accuracy as the metric, which measures the correspondence between the style factors in the output speech and those in the prompts. The accuracy of pitch, speaking speed, and volume is calculated using signal processing tools. We

fine-tune the official version of the Emotion2vec model (Ma et al., 2023) on the emotional dataset of VccmDataset, and compute the speech emotion classification accuracy with the fine-tuned model. To evaluate timbre similarity (Spk-sv) between the original prompt and the synthesized speech, we utilize the base-plus-sv version of WavLM (Chen et al., 2022). For Word Error Rate (WER), we use an ASR model [2] to transcribe the synthesized speech. This ASR model is a CTC-based HuBERT pre-trained on Librilight and fine-tuned on the 960 hours training set of LibriSpeech. For subjective evaluations, we conduct mean opinion score (MOS) evaluations on the test set to measure audio naturalness via crowdsourcing. We randomly select 30 samples from the test set of each dataset for subjective evaluation, and each audio sample is listened by at least 10 testers. We analyze the MOS in two aspects: MOS-Q (Quality, assessing clarity and naturalness of the duration and pitch) and MOS-S (Speaker similarity).

Furthermore, for the evaluation of style-controllable many-to-many scenarios in the **test set D**, we design new subjective MOS metrics: MOS-TS (Timbre Similarity), MOS-SD (Style Diversity), and MOS-SA (Style Accuracy). Specifically, the MOS-TS metric is used to assess whether the timbre remains stable across 60 different style descriptions for four speakers on the **test set D1**. The MOS-SA and MOS-SD metrics represent the accuracy and diversity of style control for each style description respectively on the **test set D2**.

## F    TRAINING AND INFERENCE SETTINGS

ControlSpeech is trained on VccmDataset using 8 NVIDIA A100 40G GPUs with each batch accommodating 3500 frames of the discrete codec. We optimize the models using the AdamW optimizer with parameters $\beta_1 = 0.9$ and $\beta_2 = 0.95$. The learning rate is warmed up for the first 5k updates, reaching a peak of $5 \times 10^{-4}$, and then linearly decayed. We utilize the open-source FACodec's voice conversion version as the codec encoder and decoder for ControlSpeech. The style-controllable baseline models are trained on the same VccmDataset training set to eliminate potential biases. We utilize a pre-trained BERT (Devlin et al., 2018) model consisting of 12 hidden layers with 110M parameters. For the implementation of the basic MDN network model, we largely follow the approach described in (Duan, 2019).

## G    MODEL ARCHITECTURE IN CONTROLSPEECH

Following (Ju et al., 2024), the basic architecture of codec encoder and codec decoder follows (Kumar et al., 2024) and employs the SnakeBeta activation function (Lee et al., 2022a). The timbre extractor consists of several conformer (Gulati et al., 2020) blocks. We use $N_{q_c} = 2$, $N_{q_p} = 1$, $N_{q_d} = 3$ as the number of quantizers for each of the three FVQ $Q^c$, $Q^p$, $Q^d$, the codebook size for all the quantizers is 1024. Text encoder and variance adaptor share the similar architecture which comprises several FFT blocks or attention layers as used by FastSpeech2 (Ren et al., 2020). The Style Extractor is a module comprising both convolutional and LSTM networks from FACodec (Ju et al., 2024) and outputs a 512-dimensional global ground truth style vector. The codec generator is a decoder primarily based on conformer blocks (Gulati et al., 2020), similar to MobileSpeech (Ji et al., 2024c). However, we opt for fewer decoder layers (6 layers) and a smaller parameter count in the codec generator.

## H    ABLATION EXPERIMENTS ABOUT MIXED DISTRIBUTIONS

In this section, we investigate **the impact of the number of mixtures in the SMSD module on model performance**. We conduct ablation studies under the *isotropic across clusters* noise perturbation mode (the mode selected for ControlSpeech), examining the effects of using 3, 5, and 7 mixtures. As shown in Table 6, the differences in the MOS-SD metric are negligible. However, an increase in the number of mixtures leads to a decline in the MOS-SA metric, indicating that an excessive number of mixtures may reduce the model's control accuracy.

---

[2] https://huggingface.co/facebook/hubert-large-ls960-ft

Table 6: Under the *Isotropic across clusters* noise perturbation scheme, we investigate the influence of the number of Gaussian mixture components in the SMSD module on stylistic diversity. Subsequently, we analyze the corresponding outcomes using the MOS-SA and MOS-SD metrics.

| Model | MOS-SA↑ | MOS-SD↑ |
|---|---|---|
| ControlSpeech w/ Isotropic across clusters w/ components=3 | 3.83±0.14 | 3.95±0.12 |
| ControlSpeech w/ Isotropic across clusters w/ components=5 | **3.84±0.12** | **4.05±0.09** |
| ControlSpeech w/ Isotropic across clusters w/ components=7 | 3.73±0.11 | 3.98±0.09 |

## I    ABLATION EXPERIMENTS ON VARIOUS NOISE MODES

We analyze the impact of different noise perturbation modes on the many-to-many style control problem, with the number of mixture distributions fixed at 5. As shown in Table 7, we find that the noise perturbation mode maintaining isotropy at the cluster centers achieves a balance between the MOS-SA and MOS-SD metrics and outperforms all other modes.

Table 7: The results of different noise perturbation modes on the MOS-SA and MOS-SD metrics.

| Model | MOS-SA↑ | MOS-SD↑ |
|---|---|---|
| ControlSpeech w/ Fully factored | 3.77±0.14 | 3.96±0.09 |
| ControlSpeech w/ Isotropic | 3.75±0.11 | 4.03±0.10 |
| ControlSpeech w/ Isotropic across clusters | **3.84±0.12** | **4.05±0.09** |
| ControlSpeech w/ Fixed isotropic | 3.72±0.13 | 3.87±0.11 |

## J    LIMITATION AND FUTURE WORK

In this work, we introduce ControlSpeech, the first TTS system capable of simultaneously cloning timbre and controlling style. While ControlSpeech has demonstrated competitive controllability and cloning capabilities, there remains considerable scope for further research and improvement based on the current framework.

**Larger Training Datasets.**    The field of style-controllable TTS urgently demands larger training datasets. Although TextrolSpeech and our VccmDataset have established a foundation, we hypothesize that achieving more advanced speech controllability may require datasets comprising tens of thousands of hours of speech with style descriptions.

**Exploring Generative Models.**    In this work, we experiment with decoupled codecs and non-autoregressive parallel generative models. In future research, we plan to explore a broader range of generative model architectures and audio representations.

