# OpenReview forum: "ControlSpeech: Towards Simultaneous Zero-shot Speaker Cloning and Zero-shot Language Style Control"
_ICLR.cc/2025/Conference — ICLR 2025 Conference Withdrawn Submission_

### Official Review · Reviewer_6TU7 · 2024-10-20

**Soundness:** 2
**Presentation:** 2
**Contribution:** 3
**Rating:** 5
**Confidence:** 5

**Summary:**

This paper presents an innovative zero-shot TTS system that allows for the control of speaking styles through textual prompts. Specifically, it adopts a pretrained decoupling codec to generate different speech attributes based on content prompts, timbre prompts, and style prompts in the corresponding representation space . Moreover, it introduces a style mixture semantic density (SMSD) module to mitigate the many-to-many mapping issue in style control. The experimental results demonstrate the proposed method exhibits comparable or state-of-the-art performance in timbre cloning and style control tasks. In addition, both the dataset and the model will be open-sourced.

**Strengths:**

1. The proposed method achieves comparable or state-of-the-art performance in timbre cloning and style control tasks.

2. The proposed SMSD and noise perturbation module effectively alleviate one-to-many issues in style control tasks

3. The dataset and model will be open-sourced.

**Weaknesses:**

The primary weakness is its reliance on label-generated style descriptions for both training and testing, which hinders the assessment of the system's robustness against free-form style prompts.

Other weaknesses include:

1. Incomplete experimental validation:

  - Missing gender accuracy metrics in style control tasks.
  - Considering the use of FACodec, it might be beneficial to include NaturalSpeech 3 as a baseline for voice cloning tasks.
  - The paper claims that each Gaussian distribution in the mixture density networks represents a specific speaking style (section 3.3, L271), but this lacks textual or experimental support.

2. Inadequate description of experimental details (refer to the question for specifics).

3. Presentation issues:

  - Related work should be presented in the main text, not the appendix.
  - In section 3.1, L158 states "the dashed box represents frame-level features," but the text encoder's output should be phoneme-level, not frame-level.
  - In section 3.2.2, L208 mentions "the aligned text representations," which is ambiguous regarding whether "text" refers to content prompts or style prompts.
  - The paper overclaims in certain instances, such as stating "ControlSpeech is the first TTS model capable of simultaneously performing zero-shot timbre cloning and style control." In fact, AudioBox supports textual style prompts , and NaturalSpeech 3 supports audio style prompts.
  - In Figure 1, the input of the content encoder and style encoder within FACodec should be audios rather than texts.
  - In Figure 2(a), the SMSD Module is labeled both as frozen and trainable, which can lead to confusion.

4. Demo issues:

  - Using ground-truth wavs as voice prompts in "Style control for unseen speakers" and "Control of unseen styles" risks leaking style information.

**Questions:**

- How is the style exactor optimized within the SMSD module, and does L_SMSD update the style exactor?

- Considering global style representations are in R^d, why use Q-K-V attention modules instead of conditional normalization layers for fusion? Additionally, can global style representations be derived from audio style prompts for testing?

- How to obtain style prompts for timbre cloning tasks?

- Is it possible to provide demos showcasing the system's performance on free-form style prompts rather than label-generated style prompts?

---

### Official Review · Reviewer_BJop · 2024-10-21

**Soundness:** 2
**Presentation:** 3
**Contribution:** 2
**Rating:** 3
**Confidence:** 4

**Summary:**

The paper introduces ControlSpeech, a text-to-speech system that can clone a speaker's voice and control speaking style with just a short audio prompt and style description. ControlSpeech uses mask-based parallel decoding and proposes the SMSD module for better style control. A toolkit is provided for validation. The system shows comparable or state-of-the-art performance, and ablation studies confirm its components' necessity.

**Strengths:**

1. ControlSpeech can simultaneously control speaker timbre and speaking style.
2. The author proposes the SMSD module to solve the many-to-many mapping problem in textual style control.
3. The paper is clearly written

**Weaknesses:**

1. The paper mentions that ControlSpeech is the first model that can control timbre and style simultaneously. However, Meta's paper AudioBox in 2023 can already achieve control of these two aspects. This is an act of overclaiming.
2. The samples from the "One timber with multiple styles" section on the demo page exhibit only minimal differences among different styles. This makes the style control ability claimed in the paper less persuasive.

**Questions:**

1. Can FACode ensure complete decoupling between content, style, and timbre?
2. The authors should add AudioBox to the baseline systems.

---

### Official Review · Reviewer_cXfJ · 2024-10-28

**Soundness:** 3
**Presentation:** 3
**Contribution:** 3
**Rating:** 5
**Confidence:** 4

**Summary:**

This paper presents a novel model, ControlSpeech, which enables independent and simultaneous control over timbre, content, and style attributes, demonstrating strong zero-shot voice cloning and style control capabilities. Additionally, the authors introduce a Style Mixture Semantic Density Sampling (SMSDS) method to address the many-to-many challenge in controllable speech generation.

**Strengths:**

1. This paper tackles a critical issue in style control by enabling independent modification of each attribute without affecting the others.
2. The introduction of the Style Mixture Semantic Density Sampling method, along with an analysis of noise perturbation, effectively addresses the many-to-many challenge in controllable speech generation.
3. The paper is well-organized, easy to follow, and includes clear and informative figures.

**Weaknesses:**

1. The disentanglement mechanism seems to be largely based on the FACodec. It would be beneficial to provide more detailed comparisons with NaturalSpeech3 to clarify the distinct contributions of this work.
2, A dedicated Related Work section would help contextualize this work by providing a clearer comparison with previous approaches, such as PromptTTS2, rather than relying solely on the brief introduction in Section 3. Please do not move the related work section into appendix.
3. Equations should be formatted using LaTeX for better readability and precision, rather than presented as plain text.
4. Additional experiments are needed to verify the independent modification of each attribute. For instance, modifying only the speech rate while evaluating speaker similarity could help determine if timbre remains unaffected.

**Questions:**

1. Does the disentanglement primarily stem from the codec generator, the codec itself, or another module?
2. Is the proposed method capable of controlling other attributes, such as age and gender, similar to PromptTTS? I do not notice any evaluation metrics based on these attributes.
3. How does ControlSpeech handle cases where there is a contradiction between the style text prompt and speaker timbre prompt?

---

### Official Review · Reviewer_FWHo · 2024-10-30

**Soundness:** 3
**Presentation:** 3
**Contribution:** 2
**Rating:** 5
**Confidence:** 4

**Summary:**

ControlSpeech achieves both zero-shot speaker cloning and zero-shot style control. Unlike previous TTS models that either mimic a speaker's voice without style control or control style without speaker-specific voice generation, ControlSpeech can independently control timbre, content, and style. It leverages a few seconds of audio prompt and a simple textual style description to fully clone a speaker's voice and adjust their speaking style. The system employs bidirectional attention, mask-based parallel decoding, and a Style Mixture Semantic Density (SMSD) module to address the many-to-many problem in textual style control.

**Strengths:**

1. Simultaneous Control: ControlSpeech's ability to simultaneously clone a speaker's voice and control style is an advancement in the TTS field.
2. Zero-Shot Capabilities: It demonstrates competitive zero-shot voice cloning and style control, which are valuable for applications where training data for specific speakers or styles is limited.
3. Disentangled Representation: By disentangling timbre, content, and style, ControlSpeech allows for more flexible and independent control over speech attributes.
4. SMSD Module: The novel SMSD module effectively addresses the many-to-many problem in style control, enhancing both style accuracy and diversity.

**Weaknesses:**

1. Insufficient innovation: There are numerous related works on zero-shot TTS and style controllable TTS. ControlSpeech combines the two tasks. The architecture used is also based on zero-shot TTS. The innovation of the method and architecture is average.
2. There is no sufficient analysis or proof of the decoupling effect of style control with or without speech prompt. As there is style information in the speech prompt. In other words, does zero-shot TTS have the ability of style control itself? Is there any test of the style control ability of zero-shot TTS? At the same time, ControlSpeech accepts speech prompt and style control at the same time. The final control effect of the two types of information on the style is not analyzed. Will there be information leakage (that is, the style in the speech prompt will also affect the final style)? In addition, in the timbre cloning experiment, the timbre similarity of ControlSpeech is not optimal, and in the style controllability experiment comparison, the pitch control effect of ControlSpeech is not optimal. Is this also because style and timbre are not completely decoupled and affect each other?

**Questions:**

1. The core innovation of the paper is to use speech prompt and style prompt to control the timbre and style of synthesized speech simultaneously. For timbre cloning, the success of zero-shot TTS has made timbre cloning reach a high degree of similarity. For style control, some text style-guided style control methods have also been proposed. It is not difficult to combine the two, so the core is how to make the synthesized audio not have a style similar to the prompt audio, but have a style that is strongly related to the style prompt. The examples provided in the "One timber with multiple styles" section of the Demo page are particularly critical. However, the demo reflects a certain diversity of styles but does not reflect the obvious controllability of the style, that is, the correlation between the style and the style prompt. Samples 1-3 reflect the change of style, while samples 4-6 cannot hear the difference in style, and the audio style of 7-9 is not very related to the style described in the text. Therefore, is the style control, the core innovation of the paper, still not well resolved?
2. Can you add experimental explanations on the correlation between the style representation of the synthesized audio and the style representation of the prompt audio, as well as the similarity between the style representation of the synthesized audio and the style representation obtained by sampling the style prompt through the SMSD module? This can help to prove that the style of the synthesized audio is more controlled by the style prompt rather than leaked by the prompt speech. For example, using different styles of speech of the same speaker as the prompt, and then using the same style prompt. Then test whether the style of the synthesized speech is consistent with the style prompt.
3. The decoupling of content and style is achieved through FACodec in Naturalspeech 3 The codec used is Ycodec = concat(Ys, Yc), where Ys=concat(Yp, Ya). How are Yp, Ya, and Yc arranged? The information used to predict the i-th layer token is the cross-attention fusion of the previous i-1 layer tokens, the text and the global style feature, and the unmasked tokens of the i-th layer. Does the order of Yp, Ya, and Yc affect the prediction effect of the token? Will the unmasked Ys in the speech prompt cause leakage and affect the predicted Ys? Thus causing the style of the generated speech to be related to the prompt audio and reduce the correlation with the style prompt?
4. This paper proposes a module called SMSD to solve the one-to-many problem of style control. The Ground Truth used by the training target of SMSD is Ys, which is extracted by the style extractor. Is the style extractor here trainable? What is the specific structure? Extracting the style of a long speech as a global style representation, is it not enough to represent more complex and style-changing prompts? Such as "a woman starts to speak softly in a low-energy voice, and then becomes more and more emotional. Her voice gradually becomes higher and higher, and her speaking speed becomes faster and faster." Sampling from different gauss distributions and noise perturbation modules increases the diversity of styles. But will the consistency of the style of the synthesized speech with the style prompt be negatively affected? If we use the same style prompt to sample multiple times and synthesize speech. How about the similarity of the global style representation extracted from this synthesized speech?

---

### Official Review · Reviewer_m7fz · 2024-11-01

**Soundness:** 4
**Presentation:** 4
**Contribution:** 4
**Rating:** 8
**Confidence:** 4

**Summary:**

The paper introduces ControlSpeech, a novel text-to-speech (TTS) system that achieves simultaneous zero-shot speaker cloning and zero-shot language style control. Unlike previous zero-shot TTS models that can clone a speaker's voice but lack style control, and controllable TTS models that can adjust speaking styles but cannot perform speaker-specific voice generation, ControlSpeech integrates both capabilities. It takes a speech prompt, a content prompt, and a style prompt as inputs, and employs bidirectional attention and mask-based parallel decoding to capture codec representations corresponding to timbre, content, and style within a discrete decoupling codec space.

The authors identify a many-to-many problem in textual style control, where different textual descriptions can correspond to the same audio style and vice versa. To address this, they propose the Style Mixture Semantic Density (SMSD) module based on Gaussian mixture density networks. The SMSD module enhances fine-grained partitioning and sampling of style semantic information, enabling more diverse and accurate speech style generation.

To evaluate ControlSpeech, the authors create a new dataset called VccmDataset and develop a toolkit named ControlToolkit, which includes source code, the dataset, and replicated baseline models for fair comparison. Experimental results demonstrate that ControlSpeech achieves comparable or state-of-the-art performance in terms of style controllability, timbre similarity, audio quality, robustness, and generalizability. Ablation studies confirm the necessity of each component in the system.

**Strengths:**

The paper makes significant contributions to the field of speech synthesis:

-> Proposes ControlSpeech, the first TTS system capable of simultaneous zero-shot speaker cloning and zero-shot style control.
-> Introduces the SMSD module to address the many-to-many problem in textual style control, enhancing style diversity and accuracy.
-> Develops the VccmDataset and ControlToolkit, providing valuable resources for the research community.
-> Demonstrates through extensive experiments that ControlSpeech achieves state-of-the-art performance in several metrics.

Major strengths:

1. Novelty: The integration of zero-shot speaker cloning with zero-shot style control addresses a significant gap in current TTS systems.
2. Technical Depth: The use of a pre-trained disentangled codec space and the SMSD module shows a deep understanding of the challenges in TTS.
3. Comprehensive Evaluation: The experiments cover a wide range of metrics and scenarios, including out-of-domain tests and ablation studies.
4. Resource Contribution: Providing the VccmDataset and ControlToolkit enhances reproducibility and aids future research.

**Weaknesses:**

1. The paper compares ControlSpeech with several baselines but could include more recent models, especially in multilingual settings or other languages beyond English.
2. While the VccmDataset is a valuable contribution, it may still be limited in scale compared to the datasets used in large-scale TTS systems. This might affect the generalizability of the results.

**Questions:**

Scalability and Data Requirements: Have you explored how ControlSpeech performs with larger datasets or in low-resource settings? What are the minimum data requirements to achieve satisfactory performance?

---

### Note · Authors · 2024-12-13

I have read and agree with the venue's withdrawal policy on behalf of myself and my co-authors.